# Deep Feature Pyramid Hashing for Efficient Image Retrieval

Adil Redaoui [†] and Kamel Belloulata *,[†]

Telecommunications Department, RCAM Laboratory, Sidi Bel Abbes University, Sidi Bel Abbes 22000, Algeria
* Correspondence: k_belloula@yahoo.fr
† These authors contributed equally to this work.

**Abstract:** Thanks to the success of deep learning, deep hashing has recently evolved as a leading method for large-scale image retrieval. Most existing hashing methods use the last layer to extract semantic information from the input image. However, these methods have deficiencies because semantic features extracted from the last layer lack local information, which might impact the global system's performance. To this end, a Deep Feature Pyramid Hashing DFPH is proposed in this study, which can fully utilize images' multi-level visual and semantic information. Our architecture applies a new feature pyramid network designed for deep hashing to the VGG-19 model, so the model becomes able to learn the hash codes from various feature scales and then fuse them to create final binary hash codes. The experimental results performed on two widely used image retrieval datasets demonstrate the superiority of our method.

**Keywords:** deep learning; deep supervised hashing; content-based image retrieval; feature pyramid; multi-scale feature





## 1. Introduction

A vast amount of data, including images and videos, is produced daily as a result of social media's fast growth. There have been many methods suggested to retrieve them effectively [1,2]. Approximate nearest neighbor (ANN) search [3] has recently drawn more and more interest due to its low computing cost and high retrieval accuracy. Among several ANN search techniques, the hashing [4] method is one of the more promising techniques to learn compact binary codes for high-dimensional input data and retrieve it at the Hamming Space.

For decades, several hashing techniques have been studied and proposed. Due to the tremendous success of deep learning, deep neural networks have been used in hashing algorithms for end-to-end learning hash codes and feature representations [5–8]. These techniques have attained cutting-edge retrieval performance, demonstrating the importance of concurrently generating similarity-preserving representations and reducing quantization error while transforming the representations into hash codes [6,9–11].

However, most of the existing deep hashing methods [12–15] used the top-layer feature to learn binary codes while disregarding down-layer features. The lower ConvLayer is generally responsible for extracting the image's visual details, including its edge, color and texture information [16]. Meanwhile, this visual information is combined in the middle layers to discover ingrained patterns. The top layer is concerned with using semantic information to produce discriminated representation [17].

Furthermore, recent research in Feature pyramids [18,19], which boost the performance by utilizing different ConvLayer in the convolutional network, has demonstrated the superiority of learning local features in a vast number of visual tasks, such as object detection [20] and image segmentation [21]. As for deep hashing, it is still full of challenges in encoding the feature pyramids into efficient binary codes.

Some of the earlier techniques functioned by feeding the network to take an image pyramid as the input (various sizes of the input image). Doing this improves the feature extraction process but increases the processing time and is not as efficient.

Even though Feature Pyramid Network (FPN) was created for object detection, we offer a novel model that uses FPN for deep hashing in this work. Using FPN makes extracting the input image's multi-level visual and semantic information easier. More specifically, an FPN creates a hierarchy of bottom-up and top-down feature maps with lateral connections of features produced from the network at various scales. We learn hash codes from these feature scales, which we fuse to obtain the final hash codes. The main contributions of this work can be summarized as follows:

1.  A Deep Feature Pyramid Hashing is proposed in this work which can fully exploit the multi-level visual and semantic information of images. Our architecture applies a new feature pyramid network designed for deep hashing to the VGG-19 model. Thus, the model becomes able to learn binary hash codes from various feature scales and then fuse them to obtain the final hash codes.
2.  To the best of our knowledge, the proposed DFPH is the first feature pyramid network-based method which generates hash codes from multiple feature scales for image retrieval.

## 2. Related Works

Thanks to their small storage capacity and quick processing speed, hashing methods have recently become widespread in image retrieval [22,23]. The primary goal of hashing is to convert high-dimensional input data into low-dimensional hash codes, with the Hamming distance reduced in similar pairs and maximizing in dissimilar pairs.

Existing hashing methods can be divided into supervised methods [24,25] and unsupervised methods [26–30] , depending on whether or not supervised data are used. Unsupervised hashing approaches [31–34] aim to learn the hash functions with unlabeled training samples, which convert input images into binary code. LSH [35] has been the most representative of the approaches. However, several other unsupervised hashing methods, such as SH [32], and ITQ [33], have been used in subsequent studies.

Unlike unsupervised hashing methods, supervised hashing techniques take supervised information from the labeled input data to learn hash code, which outperforms unsupervised approaches in terms of accuracy. Supervised Hashing with Kernels (KSH) [36] is a suggested supervised hashing method for generating a nonlinear hash function in kernel space. Unlike previous hashing approaches, Minimal Loss Hashing (MLH) [37] uses structured SVM to develop an objective function for hash function learning. Supervised Discrete Hashing (SDH) [27] provides high-quality hash codes without relaxing by reformulating the goal function.

With the rapid advancement of deep neural networks, several deep hashing algorithms [10,38–44] have recently been presented. Deep hashing can outperform approaches based on hand-crafted features because of the rich feature representations offered by deep neural networks. Pairwise and triplet-wise similarity preservation are the most prevalent ways of utilizing the relationship between labeled data when it comes to the utilization of label information.

CNNH [38] learns the hashing codes using features obtained from CNN. However, the hash function and feature representation are learned independently. There are many literature researching the hashing algorithm that the hash function and feature representation are learned independently [45,46]. Moreover, in this case, the learning hash function cannot provide feedback to a feature representation learning. Deep Pairwise-supervised Hashing (DPSH) [10] uses the Bayesian framework to create a relationship between hash codes and pairwise labels and then optimizes the relationship to learn hash functions. HashGAN [39] uses Wasserstein GAN to increase training data by exploiting pairwise similarity (dissimilarity) information, and hash codes are generated inside a Bayesian framework. In order to learn the hashing functions, Zhuang et al. [40] developed a binary CNN classifier that uses triplet-based loss for maintain semantic links. Deep Triplet Quantization (DTQ) [41] combines triplet quantization strategy in a supervised deep learning framework to enable optimization of quantization and feature learning simultaneously.

Supervised Learning of Semantics-Preserving Hash (SSDH) [42] involves building hash functions as a new fully connected layer (FC Layers), and the hash codes are learned by optimizing the cost function-specified classification error. Wang et al. [43] provided a general framework for distance-preserving linear hashing containing a deep hashing approach. In this approach, the features of the fully connected layer (FC Layers) were used for the learning hashing. Similarity-Adaptive Deep Hashing (SADH), invented by Shen et al. [44], is a two-step hashing algorithm. The fully connected layer (FC Layers) output representations aid in updating the similarity graph matrix and are then used to improve the hash code optimization process.

The approaches above use one type of feature, mostly features from the last FC layer (high-level features). However, different types of features need to be extracted to create a more complete description. Many methods have been proposed for multi-level image retrieval. Lin et al. proposed a DDH [47] method for image retrieving. The authors combine end-to-end learning, divide-and-encode, and hash code learning into a single framework. Specifically, the stack of conv-pool (convolution pooling) layers was then used to acquire multi-scale features. By combining the third pooling layer's and fourth Conv layer's outputs. In [48], Yang et al. proposed a Feature Pyramid Hashing (FPH) as a two-pyramids (vertical and horizontal) image hashing architecture to learn the subtle appearance details and the semantic information for fine-grained image retrieval. Ng et al. [49] developed a novel multi-level supervised hashing (MLSH) technique for image retrieval. The authors built and trained the tables individually, utilizing different levels of features (semantic and structural).

### 3. Proposed Method

This section will detail the suggested Deep Feature Pyramid Hashing for Image Retrieval. We begin by presenting the problem definition for learning hash codes. Then, we introduce the model architecture. Finally, we give our proposed method's objective function.

#### 3.1. Problem Definition

Let $X = \{x_i\}_{i=1}^N \in \mathbb{R}^{d \times N}$ denote the training dataset with $N$ image samples, and $d$ features sizes. Let $Y = \{y_i\}_{i=1}^N \in \mathbb{R}^{K \times N}$ represent the ground truth labels of the training dataset $X$, where $K$ is the number of classes.

Let $S = \{s_{ij}\}$ be the matrix to denote the pair-wise labels for training samples, with $s_{ij} \in \{0, 1\}$, where the element $s_{ij} = 1$ represents that $x_i$ and $x_j$ samples are semantically similar, $s_{ij} = 0$ represents that $x_i$ and $x_j$ samples are not semantically similar. The goal of deep hashing techniques with pair-wise labels is to learn a nonlinear deep hash function $f : x \mapsto B \in \{-1, 1\}^L$, which can encode each sample $x_i$ into binary codes $b_i \in \{-1, 1\}^L$, $L$ is the length of hash codes.

#### 3.2. Model Architecture

Feature pyramid network (FPN) was proposed by Lin [19] as a feature extractor design. FPN generates multiple feature map layers (multi-scale feature maps) with better-quality information than the regular feature pyramid.

Figure 1 shows the network's architecture of the proposed deep hashing for image retrieval. The bottom-up hierarchy uses VGG-19 to construct the bottom-up hierarchy. Specifically, we use the feature output of the last convolutional layer in each convolutional block ($conv3 - 5$). We denote the output of these last convolutional layers as $C3, C4, C5$ for $conv3, conv4,$ and $conv5$ outputs, respectively. We omit conv1 and conv2 into the pyramid due to its large memory footprint. Table 1 details the convolutional layers' parameters and feature sizes of different convolutional blocks.

**Table 1.** Details of the feature extraction network. Note that we use the features of the layers marked by "#". For simplicity, we omit ReLU and Batch Normalization layers.

| Conv Block | Layers | Kernel Size | Feature Size |
|:---:|:---:|:---:|:---:|
| 1 | Conv2D<br>Conv2D #<br>MaxPooling | $64 \times 3 \times 3$<br>$64 \times 3 \times 3$<br>$2 \times 2$ | $224 \times 224$ |
| 2 | Conv2D<br>Conv2D #<br>MaxPooling | $128 \times 3 \times 3$<br>$128 \times 3 \times 3$<br>$2 \times 2$ | $112 \times 112$ |
| 3 | Conv2D<br>Conv2D<br>Conv2D<br>Conv2D #<br>MaxPooling | $256 \times 3 \times 3$<br>$256 \times 3 \times 3$<br>$256 \times 3 \times 3$<br>$256 \times 3 \times 3$<br>$2 \times 2$ | $56 \times 56$ |
| 4 | Conv2D<br>Conv2D<br>Conv2D<br>Conv2D #<br>MaxPooling | $256 \times 3 \times 3$<br>$256 \times 3 \times 3$<br>$256 \times 3 \times 3$<br>$256 \times 3 \times 3$<br>$2 \times 2$ | $28 \times 28$ |
| 5 | Conv2D<br>Conv2D<br>Conv2D<br>Conv2D #<br>MaxPooling | $256 \times 3 \times 3$<br>$256 \times 3 \times 3$<br>$256 \times 3 \times 3$<br>$256 \times 3 \times 3$<br>$2 \times 2$ | $14 \times 14$ |

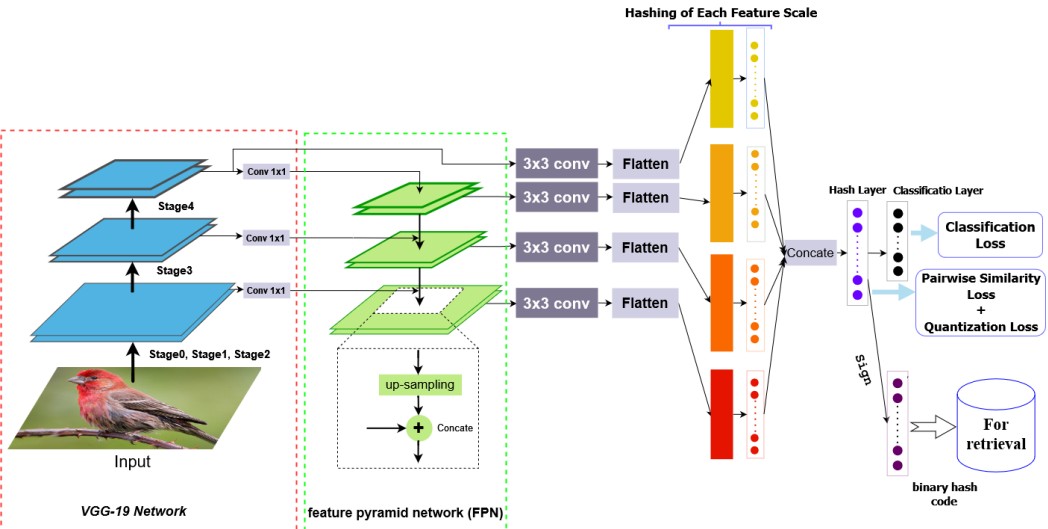

**Figure 1.** This figure shows our proposed model: Deep Feature Pyramid Hashing (DFPH), which uses VGG19 as the backbone and uses the feature pyramid network (FPN) and the created layers for hashing.

As we go down the top-down path, we upsample the previous layer using bilinear interpolation. Moreover, we apply a $1 \times 1$ convolution filter (reduce *C3*, *C4*, and *C5* channels) to the corresponding feature maps in the bottom-up pathway. Then, we add them element-wise.

FPN extracts four final features (multi-scale feature maps). The bottom and upper layers are selected to exploit spatial resolution and semantic information because the spatial resolution decreases as we go up in the layers.

For the obtained features, we fuse them using a $3 \times 3$ convolutional layer to obtain a fused feature and apply dropout layers (to avoid overfitting). After that, we attach the hashing layer on this final set of feature maps to produce the first hashing codes. Note that we use the *Relu* activation function for the first hashing layers.

At the end of the architecture, we concatenated the four hashing layers. Then, we connected this layer to the final hashing layer and connected the latter with the classification layer (the sum of neurons equals the sum of classes of the dataset). With running this procedure, the network utilizes different hashing results based on various scale features. As a result, the network could retrieve the images better.

### 3.3. Objective Function

To learn similarity-preserving hash codes, we develop the following loss functions: pairwise similarity loss, pairwise quantization loss, and classification loss. All of them are merged to train our model.

Pairwise Similarity Loss. We perform our deep hashing model by preserving similarities as possible of all pairwise images in the Hamming space. The pairwise similarity is computed using the inner product. The inner product $\langle .,. \rangle$ for two binary codes, $b_i$ and $b_j$, is formulated as follows: $dist_H(b_i, b_j) = \frac{1}{2} b_i^T b_j$

Given that all of the points' binary codes are $B = \{b_i\}_{i=1}^N$ , we can formulate the likelihood of the pairwise labels $S = \{s_{ij}\}$ as:

$$p\left(s_{ij}|B\right) = \begin{cases} \sigma(w_{ij}) & s_{ij} = 1 \\ \\ 1 - \sigma(w_{ij}) & s_{ij} = 0 \end{cases} \tag{1}$$

where $\sigma(w_{ij}) = \frac{1}{1+e^{-w_{ij}}}$, and $w_{ij} = \frac{1}{2} b_i^T b_j$

Based on the equation above, we can infer that the larger the inner product $\langle b_i, b_j \rangle$, the smaller the corresponding $dist_H(b_i, b_j)$, and the larger $p(1|b_i, b_j)$. This also implies that the hash codes $b_i$ and $b_j$ are considered as similar when $s_{ij} = 1$, and vice versa.

By taking the negative log-likelihood of the observed pairwise labels in $S$, we can obtain the following optimization problem:

$$J_1 = -\log p(S|B) = -\sum_{s_{ij} \in S} (s_{ij} w_{ij} - \log(1 + e^{w_{ij}})) \tag{2}$$

The above formula (optimization problem) makes the Hamming distance between two similar (dissimilar) points to be as small (large) as possible, which is precisely what pairwise similarity-based hashing techniques aim to achieve.

Pairwise Quantization Loss. Discrete hash codes are utilized in real-world applications to determine similarity. However, discrete hash coding in CNN is challenging to optimize. Therefore, the continuous form of hash coding is used to prevent gradient disappearance during the back-propagation phase. We define the output of the hash layer as $u_i$ and make $b_i = \text{sgn}(u_i)$.

Hence, quantization loss is introduced to minimize the gap between continuous and discrete hash codes. The second objective is defined as:

$$J_2 = \sum_{i=1}^Q \parallel b_i - u_i \parallel_2^2 \tag{3}$$

where $Q$ is the size of the mini-batch.

Classification Loss. To achieve robust representations of multi-scale features during training of deep network, we apply the classification loss to find the classification of the classes. The classification loss can be expressed as follows:

$$J_3 = -\sum_{i=1}^{Q}\sum_{k=1}^{K} y_{i,k} \log(p_{i,k}) \tag{4}$$

where $y_{i,k}$ is the ground-truth label, $p_{i,k}$ is the softmax output of the $i-th$ training sample, which belongs to the $k-th$ class.

To sum up, the total loss function can be achieved by combining pairwise similarity loss, pairwise quantization loss, and classification loss:

$$\begin{aligned}
\min_{\mathbf{B},\mathbf{U},\mathbf{P}} J &= J_1 + \beta J_2 + \gamma J_3 \\
&= -\sum_{s_{ij}\in S} (s_{ij}w_{ij} - \log(1 + e^{w_{ij}})) \\
&+ \beta \sum_{i=1}^{Q} \| b_i - u_i \|_2^2 \\
&- \gamma \sum_{i=1}^{Q}\sum_{k=1}^{K} y_{i,k} \log(p_{i,k})
\end{aligned} \tag{5}$$

where $\beta$, $\gamma$ are hyper parameters that control importance of the J2 and J3 terms. The final objective function in Equation (5) is convex and can be optimized by the Adam algorithm [50]. At the training phase, J1 makes similar binary codes as close as possible and places dissimilar binary codes as far away as possible. J2 quantization loss is introduced to minimize the gap between continuous and discrete hash codes. J3 can use semantic supervision to direct how discrete hash codes are learned and add more semantic information to the hash codes that are created to improve hash code discrimination.

## 4. Experiments

We verify the effectiveness of our method on two public datasets: CIFAR-10 and NUS-WIDE. We first present these datasets in brief and then explore experimental settings. Evaluation metrics and baselines are given in Section 4.3. In the last section, we show the results with validations and comparisons to various state-of-the-art hashing methods.

### 4.1. Datasets

CIFAR-10 [51] database contains 60,000 images composed of 10 categories, each image with the size of 32 × 32. Following to [52], we randomly selected 100 images per class as the queries (1000 testings) and the rest images as the database. Furthermore, we randomly sampled 500 images per category (5000 images) from the retrieval database as the training set.

NUS-WIDE [53] is a multi-label database that contains nearly 270,000 color images collected from Flickr, and is composed of 81 labels. We randomly selected 2100 images from the 21 most frequent concepts as the query set, while the rest as the database, and randomly selected 10,000 images from the database as a training dataset.

### 4.2. Experimental Settings

Our implementation of DFPH is based on the Pytorch framework, which is based on the Torch library. We adopt a VGG-19-convolutional network, pretrained on ImageNet [54]. In all experiments, we used the Adam algorithm [50] to train our network (learning rate $= 1e^{-5}$ ). For the hyper-parameter of the loss function, alpha is set to 0.01 and beta to 0.1.

### 4.3. Evaluation Metrics and Baselines

To evaluate the retrieval performance of different approaches, we use four evaluation metrics: (1) Mean Average Precision (MAP); (2) Precision–Recall curves (PR); (3) Precision curves with respect to the number of top returned results (P@N); (4) and Precision within Hamming radius 2 (P@H ≤ 2). We compare the proposed DFPH with several of traditional or state-of-the-art hashing techniques, including five shallow unsupervised methods, i.e., LSH [3] , SGH [55] , SH [32], PCAH [56], ITQ [33], and two shallow supervised hashing methods, i.e., SDH [27] , KSH [36] , and eight deep supervised hashing methods, i.e., CNNH [38], DPH [57], DNNH [7], DHN [6], DCH [58], LRH [52], HashNet [9], DHDW [59]. For the multi-label NUS-WIDE and CIFAR-10 data-sets, if two samples share the same semantic labels, they are similar. Otherwise, they are not similar.

### 4.4. Results

The MAP results for all techniques on CIFAR-10 and NUS-WIDE datasets with various hash code lengths are presented in Tables 2 and 3. The tables show that the proposed method substantially outperforms all compared techniques. Specifically, compared to SDH, the best shallow hashing method, DFPH achieves absolute increases of 49.8% and 24.4% in average mAP scores on CIFAR-10 and NUS-WIDE datasets, respectively. However, we can see that the performance of deep hashing is better than traditional hashing methods, mainly due to its ability to create more robust representations of features. For deep hashing methods, compared to the second-best hashing method LRH, the average mAP of our method was increased by 11.5% and 8.3% on CIFAR-10 and NUS-WIDE, respectively.

**Table 2.** Mean Average Precision (MAP) of Hamming ranking for different number of bits on CIFAR-10.

| Method | CIFAR-10 (MAP) | | | |
| --- | --- | --- | --- | --- |
| | **12 Bits** | **24 Bits** | **32 Bits** | **48 Bits** |
| DFPH | 0.800 | 0.823 | 0.838 | 0.840 |
| DPH [57] | 0.698 | 0.729 | 0.749 | 0.755 |
| LRH [52] | 0.684 | 0.700 | 0.727 | 0.730 |
| HashNet [9] | 0.609 | 0.644 | 0.632 | 0.646 |
| DHN [6] | 0.555 | 0.594 | 0.603 | 0.621 |
| DNNH [7] | 0.552 | 0.566 | 0.558 | 0.581 |
| CNNH [38] | 0.439 | 0.511 | 0.509 | 0.522 |
| SDH [27] | 0.285 | 0.329 | 0.341 | 0.356 |
| KSH [36] | 0.303 | 0.337 | 0.346 | 0.356 |
| ITQ [33] | 0.162 | 0.169 | 0.172 | 0.175 |
| SH [32] | 0.127 | 0.128 | 0.126 | 0.129 |

Figures 2a and 3a show the retrieval performance in terms of precision curves (P@H = 2). The proposed model significantly outperforms other methods. The precision curves show that the proposed DFPH method still has the best precision rate when the code length increases.

**Table 3.** The MAP of DFPH and other hashing methods on NUS-WIDE dataset, MAP are calculated within the top 5000 returned images.

| Method | NUS-WIDE (MAP) | | | |
| --- | --- | --- | --- | --- |
| | **12 Bits** | **24 Bits** | **32 Bits** | **48 Bits** |
| DFPH | 0.826 | 0.850 | 0.853 | 0.859 |
| DPH [57] | 0.770 | 0.784 | 0.790 | 0.786 |
| LRH [52] | 0.726 | 0.775 | 0.774 | 0.780 |
| DHN [6] | 0.708 | 0.735 | 0.748 | 0.758 |
| HashNet [9] | 0.643 | 0.694 | 0.737 | 0.750 |
| DNNH [7] | 0.674 | 0.697 | 0.713 | 0.715 |
| CNNH [38] | 0.611 | 0.618 | 0.625 | 0.608 |
| SDH [27] | 0.568 | 0.600 | 0.608 | 0.637 |
| KSH [36] | 0.556 | 0.572 | 0.581 | 0.588 |
| ITQ [33] | 0.452 | 0.468 | 0.472 | 0.477 |
| SH [32] | 0.454 | 0.406 | 0.405 | 0.400 |

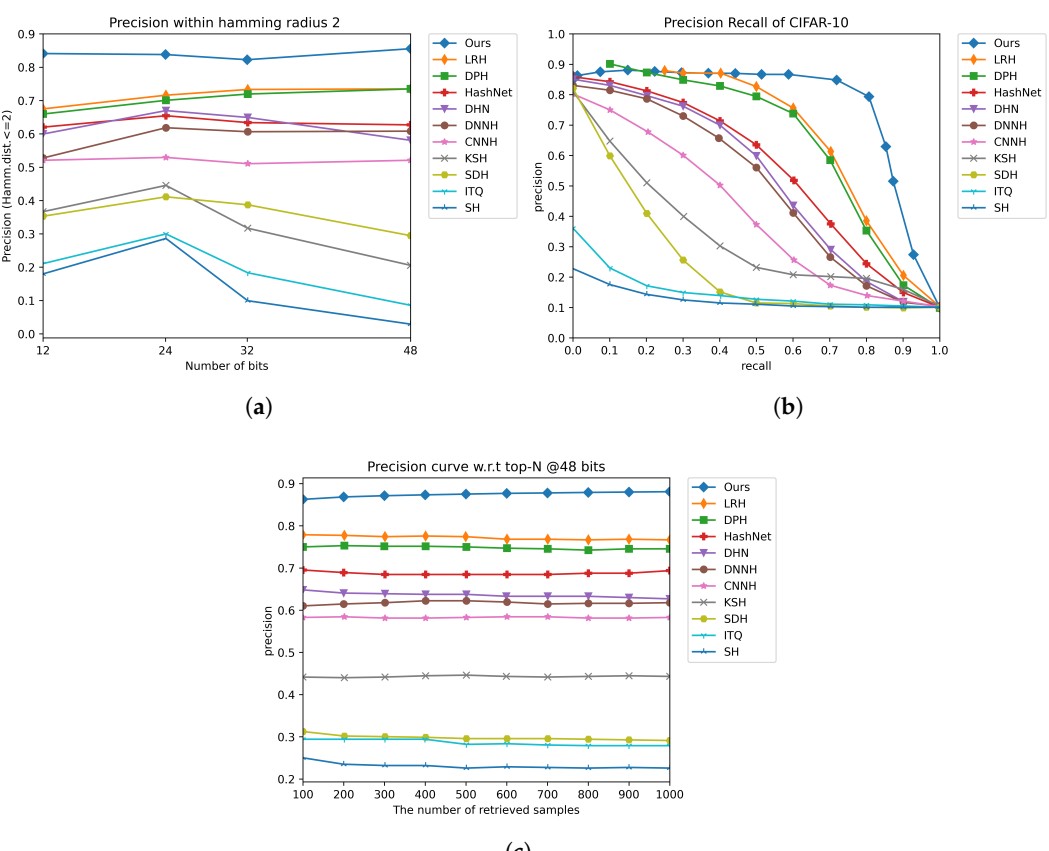

(a)

(b)

(c)

**Figure 2.** The comparison results on the CIFAR-10 dataset under three evaluation metrics. (**a**) Precision within hamming radius 2. (**b**) Precision recall curve on 48 bits. (**c**) Precision curve w.r.t top-N @48 bits.

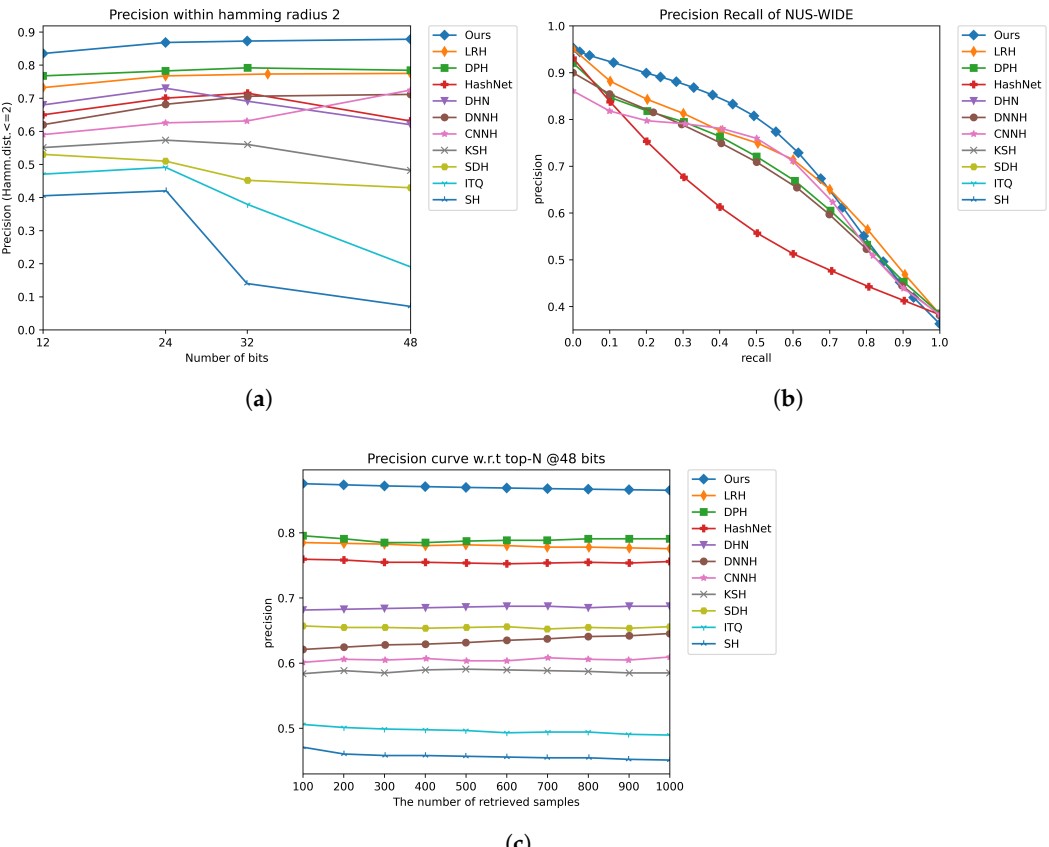

**Figure 3.** The comparison results on the NUS-WIDE dataset under three evaluation metrics. (**a**) Precision within hamming radius 2. (**b**) Precision Recall Curve on 48 bits. (**c**) Precision curve w.r.t top-N @48 bits.

We further demonstrate the efficiency of our method by comparing its precision–recall and precision with respect to top returned samples with other methods in Figures 2b,c and 3b,c, respectively. From Figures 2c and 3c, it can be seen that the proposed DFPH method achieves the best precision with 48 bits when the number of return samples is from 100 to 1000. From Figures 2b and 3b, our DFPH achieves substantially high precision at a low recall level, which is required for precision-first retrieval, and widely used in practical systems. Generally speaking, our method DFPH has better performance than the compared methods.

Moreover, the retrieval accuracy for CIFAR-10 dataset is shown for different categories of images as illustrated in Figure 4. For this example, we have proved that we can remove the irrelevant images by using our DFPH approach. Figure 4 shows the top 20 retrieved images from CIFAR-10 dataset by DFPH with 48-bit hash codes. The first column shows the query images, the retrieval results of DFPH are shown in other columns.

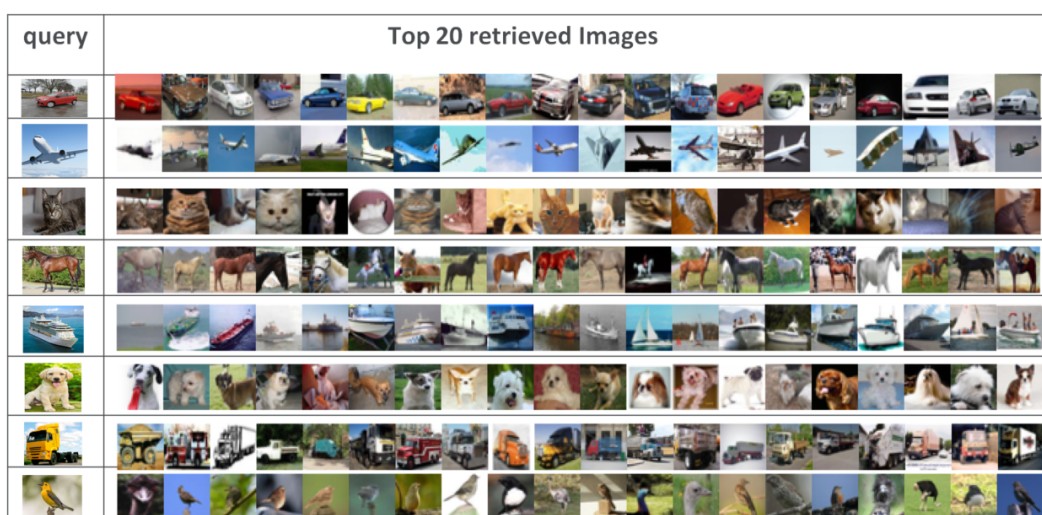

**Figure 4.** Top 20 retrieved results from CIFAR-10 dataset by DFPH with 48-bit hash codes. The first column shows the query images, the retrieval results of DFPH are shown at other columns.

*4.5. Ablation Studies*

(1)    Using various fundamental feature extractors: We replace VGG19 with VGG13 and VGG16, and Table 4 details their performance on the CIFAR-10 dataset. The table shows that employing deeper networks can enhance image retrieval performance. We thus employ VGG19 as our fundamental feature extractor.

**Table 4.** Mean Average Precision (MAP) of Hamming ranking for different number of bits using VGG13, VGG16 and VGG19 as the fundamental feature extractors on CIFAR-10 dataset. The best results are highlighted with bold.

| Method | 12 Bits | 24 Bits | 32 Bits | 48 Bits | Conv Layers Num |
|--------|---------|---------|---------|---------|-----------------|
| VGG13  | 0.759   | 0.798   | 0.782   | 0.796   | 10              |
| VGG16  | 0.763   | 0.824   | 0.824   | 0.821   | 13              |
| VGG19  | **0.800** | **0.823** | **0.838** | **0.840** | 16            |

(2)    Pyramid representations efficiency: We compare our DFPH method with its variants to further explore the effect of different scale features on performance. We use the single-scale map of C4 and C5 (Conv 4; Conv 5) and remove the top-down pathway. With this modification, the 3 × 3 convolutions connections followed by 1 × 1 convolutions are attached to the bottom-up pyramid.

Table 5 shows the retrieval performance of single-level and multi-level features on CIFAR-10. It can be seen that using a feature from 'Conv 5' achieves the most significant average mAP score of 76.5%, while that from 'Conv 4' only achieves 58%. Moreover, our DFPH has achieved the average mAP scores of 82.75%, 6.25% higher than simply using single-level features. The hash codes achieve the best performance when all scale features are.

In Figure 5, we display the precision–recall curves of DFPH in the case of various scale features. Our DFPH retains over 80% precision and nearly identical precision–recall curves at hash bits 12, 24, 36, and 48. DFPH achieves superior precision and recall with the same hash code length compared to single-level features. The binary hash codes perform best when all feature scales are used. It proves that high-level characteristics are more effective in carrying information when creating hash codes. While low-level features can contribute supplementary information to the high-level features information, low-level features cannot entirely take the place of high-level

characteristics. The information contained in each scale's features is essential. It further demonstrates how well DFPH makes use of all scales' features.

**Table 5.** Mean Average Precision (MAP) of Hamming ranking of different scales for different number of bits on CIFAR-10.

| Method | CIFAR-10 (MAP) | | | |
| | 12 Bits | 24 Bits | 32 Bits | 48 bits |
|---|---|---|---|---|
| $conv_5$ | 0.634 | 0.811 | 0.801 | 0.814 |
| $conv_4$ | 0.540 | 0.530 | 0.634 | 0.619 |
| DFPH | 0.800 | 0.827 | 0.838 | 0.845 |

(3) Ablation studies of the objective function: We evaluate the impact of Pairwise Quantization Loss and Classification Loss constraints, which reflects the effects of hash coding and classification on CIFAR-10. The experimental configuration is based on the proposed DFPH method, where $\beta$ and $\gamma$ are the relevant parameters for $J2$ and $J3$. The model is designated DFPH-J3 when $\beta = 0$ and DFPH-J2 when $\gamma = 0$. As seen in Table 6, if $\beta$ and $\gamma$ are not 0, then each term of the suggested loss function constrains the creation of hash codes, and our method has achieved a 4.5% improvement. $J2$ and $J3$ make nearly identical enhancements. The primary reason is that for the whole model, $J2$ and $J3$ play a role in reducing quantization error and semantic preservation, respectively. The performance of our model as a whole may decrease if one of them is eliminated.

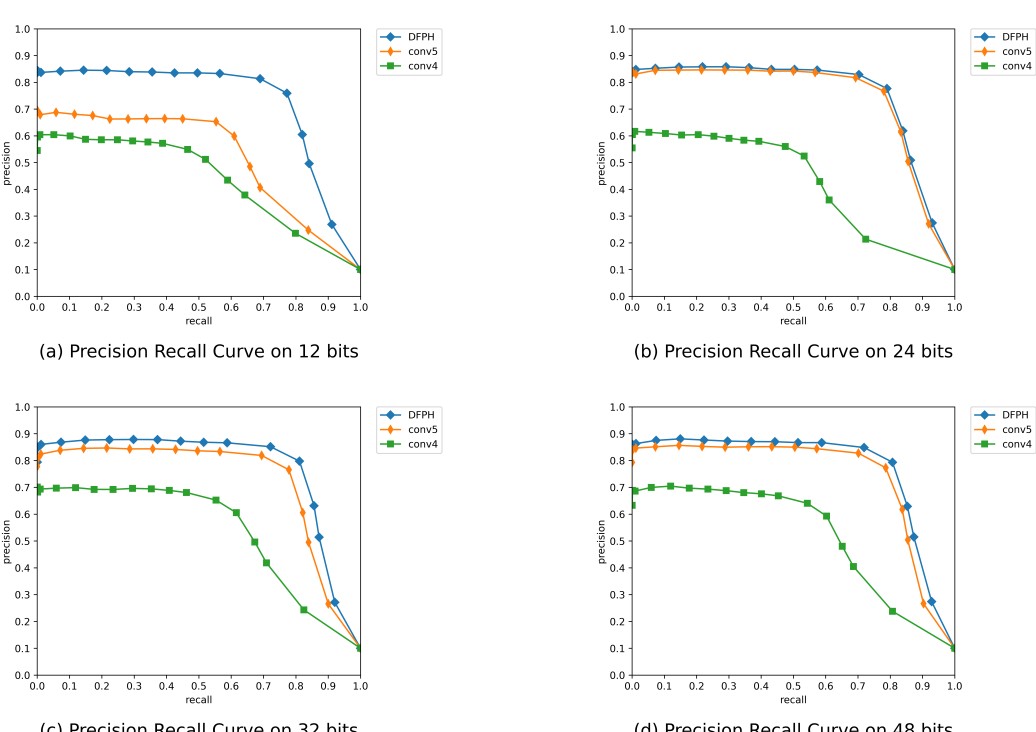

(a) Precision Recall Curve on 12 bits

(b) Precision Recall Curve on 24 bits

(c) Precision Recall Curve on 32 bits

(d) Precision Recall Curve on 48 bits

**Figure 5.** Precision–recall curve of different scales features on CIFAR-10 with hash bits 12, 24, 32 and 48.

**Table 6.** MAP values of different variants of our objective function for CIFAR-10 data set.

| | CIFAR-10 (MAP) | | | |
|---|---|---|---|---|
| **Method** | **12 Bits** | **24 Bits** | **32 Bits** | **48 Bits** |
| DFPH-J_2 | 0.721 | 0.780 | 0.809 | 0.771 |
| DFPH-J_3 | 0.771 | 0.766 | 0.783 | 0.809 |
| DFPH | 0.800 | 0.827 | 0.838 | 0.845 |

*4.6. Parameter Sensitivity Analysis*

To analyze the influence of hyper-parameters $\beta$ and $\gamma$ on retrieval performance, we conducted experiments to determine the impact of the choice of $\beta$ and $\gamma$ on retrieval performance. We tested the $\beta$ and $\gamma$ from 0.0001 to 0.1 with a multiplicative step-size ten on the CIFAR-10 dataset in the case of $48 - bit$ code. In Figure 6, we display the changes in mAP curves of DFPH concerning the variation of $\beta$ and $\gamma$. We can see that MAP initially increases with the increase of $\beta$ and $\gamma$, while the performances progressively stabilize after reaching their peak. Additionally, we can always obtain satisfactory performance in a large range of $\beta$ and $\gamma$, which shows that our method is not sensitive to parameters $\beta$ and $\gamma$ and accordingly proves the robustness of our method.

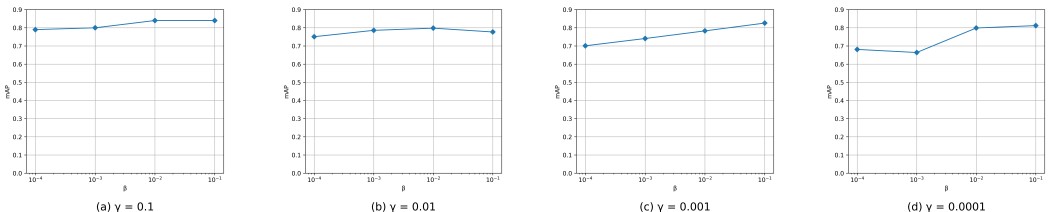

**Figure 6.** Parameter sensitivity analysis on CIFAR-10 dataset with 48-bit of the proposed DFPH.

**5. Conclusions and Future Work**

In this paper, we developed an end-to-end multi-scale deep Hashing method for large-scale image retrieval. Our architecture applies a new feature hierarchy network designed for deep hashing to generate multi-scale features. In addition, efficient binary codes are generated by learning hash codes from different feature scales and then combining them to create final binary hash codes. The experimental results on two image retrieval datasets demonstrated that our method outperforms other state-of-the-art hashing methods. This multi-level visual and semantic information can also be very useful in a non-hashing system such as detection and segmentation. In this case, to increase the detection performance of multi-scale information, we can add a novel Multi-Scale Image Fusion layer to the backbone network. In future work, it is interesting to apply our proposed framework to medical image data sets. Since medical images contain objects at different scales, especially as many of them are very small, using our method helps to increase retrieval performance.

**Author Contributions:** Conceptualization, A.R.; methodology, K.B.; software, A.R.; validation, K.B. and A.R.; formal analysis, A.R. and K.B.; investigation, K.B.; writing–original draft preparation, A.R. and K.B.; writing–review and editing, K.B. and A.R.; visualization, A.R.; supervision, K.B.; project administration, K.B.; funding acquisition, K.B. All authors have read and agreed to the published version of the manuscript.

**Funding:** This research received no external funding.

**Data Availability Statement:** Publicly available datasets were analyzed in this study. These data can be found here: http://www.cs.toronto.edu/~kriz/cifar.html    https://paperswithcode.com/datasets/nuswide (all accessed on 18 June 2021).

**Conflicts of Interest:** The authors declare no conflict of interest.

## Abbreviations

The following abbreviations are used in this manuscript:

DFPH    Deep Feature Pyramid Hashing for efficient Image Retrieval
CBIR    Content-Based Image Retrieval
FPN    Feature Pyramid Network
CNN    Convolutional Neural Network
DCNN    Deep Convolutional Neural Network

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
