# Peer review of "Deep Feature Pyramid Hashing for Efficient Image Retrieval"

_information, doi:10.3390/info14010006_

Round 1

Reviewer 1 Report

In this paper, the authors proposed a deep feature based image retrieval technique. their model is able to learn the hash codes from various feature scales and fuse them to create final hash codes. The technique is novel in nature and is effective than the existing techniques. There are some points which authors need to incorporate in their manuscript.

1. [?] in line number 29(on first page) needs to be corrected.

2. No need to include point number 3 in contributions of the work(on page number 2). 

3. Many of the abbreviations are repeated. For example- DMSH is two times repeated on page number 2. check the manuscript thoroughly for this correction.

4. Remove the paragraph - "Some of the earlier techniques functioned by feeding the network, is to take an image pyramid as the input (various sizes of the input image). Doing this indeed improves the feature extraction process but also increases the processing time and is not as efficient. " from section 2.2 and include it in Section 1 prior to Contributions of the work.

5.On line number 81-82, the authors claim that- FPN 81 extracts four final features, each presenting the input image’s features at various scales. Why four final features. Kindly explain.

6. Which library is used to implement losses in the objective functions. Explain it in Experiments section.

7. In line number 60, where Y = {yi} N i=1 ∈ RK×N represents the ground truth labels of the xi , K is the number of classes. It is Yi not Y which represents the ground truth labels. 

8. Discuss the complexity of the proposed model.

Reviewer 2 Report

In this manuscript, the authors proposed an end-to-end multi-scale deep segmentation method for hashing image retrieval. The experimental results on two image retrieval datasets demonstrated the effectiveness of the method. There are some concerns:

1. How many hours does it cost to train your hashing method?

2. An ablation study of super-parameters in the total loss is needed.

3.Can the multi-level visual and semantic information be used in a non-hashing system ?

Reviewer 3 Report

The manuscript presents the Deep Feature Pyramid Hashing (DFPH), which fully utilizes images’ multi-level visual and semantic information. A new feature pyramid network is designed for deep hashing based on the VGG-19 model. The manuscript is well written and presented. The experiments are extensive and many state-of-the-art methods are compared with the DFPH. However, I have some major comments:
1.In the line 79-80, “CNNH [38] learns the hashing codes using features obtained from CNN. However, The hash function and feature representation are learned independently”
There are many literature researching the hashing algorithm that the hash function and feature representation are learned independently. Such as the following literature [1] [2]. Please cite the two references [1] [2] to make this section completed.
[1]Y. Fang, P. Li, J. Zhang and P. Ren, "Cohesion Intensive Hash Code Book Coconstruction for Efficiently Localizing Sketch Depicted Scenes," in IEEE Transactions on Geoscience and Remote Sensing, vol. 60, pp. 1-16, 2022, Art no. 5629016, doi: 10.1109/TGRS.2021.3132296.
[2]Q. Y. Jiang, W. J. Li, Member, and IEEE, “Discrete latent factor model for cross-modal hashing,” IEEE Transactions on Image Processing, vol. PP, no. 99, 2017.
2.In the line 188, “R^{dxN}”. The parameter d is not defined.
3.The Feature Pyramid Network (FPN) is an important algorithm in this manuscript. However, FPN is not explained in detail but only cite some relevant literature. In addition, In Fig1, The architecture of FPN (stage0 -stage4) should be marked distinctly to make the paper more comprehensive for non professionals.
4.The loss functions: pairwise similarity loss, pairwise quantization loss, and classification loss should be explicit that maximization or minimization.

Round 2

Reviewer 1 Report

The paper is acceptable in current form.

Author Response

The authors should express our sincere appreciations to the reviewers for spending valuable time to review our manuscript and give kindly and detailed recommendations for the improvement of our manuscript.
We think that we have provided a point-by-point response to the concerns according to the reviewers’ reports.
We do not believe it would be appropriate to include the discussion on the complexity of the proposed approach for a number of reasons. First, there are no standard (reference) complexity algorithms. Thus, which algorithms' GPU time should we include? Secondly, all the algorithms that we refer to and were cited in the paper didn't study the complexity. Thirdly, by decoupling the network training (offline) and retrieval by query (online), one can concentrate on the complexity of the latter issue and leave the GPU efficiency to the former (the VGG-19 + FPN network).Finally, the overall complexity is composed of training complexity and online test complexity. For online complexity, the calculation of the hash codes can be performed using a few matrix multiplications, which is fast and is linear with the number of query data and irrelevant with the size of training data. The training time complexity is linear to the size of the training data.

Reviewer 3 Report

The manuscript can be accepted in this version.

Author Response

The authors should express our sincere appreciations to the reviewers for spending valuable time to review our manuscript and give kindly and detailed recommendations for the improvement of our manuscript.